# Two-Layer Rubber-Based Composite Material and UHMWPE with High Wear Resistance

**DOI:** 10.3390/ma15134678

**Published:** 2022-07-04

**Authors:** Afanasy A. Dyakonov, Andrey P. Vasilev, Sakhayana N. Danilova, Aitalina A. Okhlopkova, Praskovia N. Tarasova, Nadezhda N. Lazareva, Alexander A. Ushkanov, Aleksei G. Tuisov, Anatoly K. Kychkin, Pavel V. Vinokurov

**Affiliations:** 1Department of Chemistry, Institute of Natural Sciences, North-Eastern Federal University, 677000 Yakutsk, Russia; afonya71185@mail.ru (A.A.D.); gtvap@mail.ru (A.P.V.); okhlopkova@ya.ru (A.A.O.); pn.tarasova@mail.ru (P.N.T.); lazareva-nadia92@mail.ru (N.N.L.); alexanderushkanov@mail.ru (A.A.U.); 2Institute of the Physical-Technical Problems of the North, Siberian Branch of the Russian Academy of Sciences, 677980 Yakutsk, Russia; kychkinplasma@mail.ru; 3Federal Research Centre “The Yakut Scientific Centre of the Siberian Branch of the Russian Academy of Sciences”, 677000 Yakutsk, Russia; tuisovag@gmail.com; 4Department of Radio Physics and Electronic Systems, Institute of Physics and Technologies, North-Eastern Federal University, 677000 Yakutsk, Russia; pv.vinokurov@s-vfu.ru

**Keywords:** isoprene rubber, phase boundary, rubber, two-layer material, ultra-high-molecular-weight polyethylene

## Abstract

The aim of the study is the development of two-layer materials based on ultra-high-molecular-weight polyethylene (UHMWPE) and isoprene rubber (IR) depending on the vulcanization accelerators (2-mercaptobenzothiazole (MBT), diphenylguanidine (DPG), and tetramethylthiuram disulfide (TMTD)). The article presents the study of the influence of these accelerators on the properties and structure of UHMWPE. It is shown that the use of accelerators to modify UHMWPE leads to an increase in tensile strength of 28–53%, a relative elongation at fracture of 7–23%, and wear resistance of three times compared to the original UHMWPE. It has been determined that the introduction of selected vulcanization accelerators into UHMWPE leads to an increase in adhesion between the polymer and rubber. The study of the interfacial boundary of a two-layer material with scanning electron microscopy (SEM) and infrared spectroscopy (FTIR) showed that the structure is characterized by the presence of UHMWPE fibrils localized in the rubber material due to mechanical adhesion.

## 1. Introduction

Currently, elastomeric materials are widely used as sealing devices, damping elements, lining materials, rubber bearings, medical implants, in particular, bearing components on the total hip prosthesis, etc. [1,2,3]. Elastomers containing rubber and fillers are characterized by high resistance to fatigue and wear, absorbing vibration, and having excellent chemical stability and oil resistance, which allows them to be used for manufacturing rubber moving parts for various mechanisms [4]. However, the development of industry challenges us to search for new materials with improved performance. To improve the performance of elastomers, the following approaches are used: Volume modification, coating, creation of hybrid materials based on a combination of two different materials, etc. [5,6,7,8]. The advantages of surface modification of elastomers include the preservation of useful volumetric properties of rubber along with improvement of the surface characteristics (antifriction properties, resistance to aggressive media, and ultraviolet radiation) [9,10,11]. Coatings on the rubber surface are produced in various ways: Plasmochemical treatment of elastomer surface [12,13], ion-plasma application of a metal layer [14], application of durable and wear-resistant polymers [15,16], and production of hybrid materials [17,18].

The listed methods for applying antifriction coatings on elastomers make it possible to maintain volumetric properties with no interference with the vulcanization process. However, there is a problem due to the low adhesion strength between the coating and the elastomer material, low wear resistance and elasticity of the applied layer, as well as the difficulty of controlling the thickness of the applied material on the elastomer. Two thermodynamically incompatible polymers that do not form an equilibrium thermodynamic system are considered unable to form a strong connection with each other. Previously, we showed that it is possible to form a strong adhesive interaction between a thermoplastic and an elastomer due to the formation of chemical bonds at the interphase in the presence of certain vulcanization accelerators. In this regard, the determination of the regularities of the vulcanization accelerators’ influence on the interaction between thermoplastics and elastomers will expand the technological possibilities of creating various combinations of materials based on polymers of different chemical nature. Further development of surface modification techniques of rubber resulted in our use of UHMWPE for these purposes. UHMWPE polymer as a structural material has high toughness, high wear/abrasion resistance, impact resistance, chemical inertness to corrosive media and low coefficient of friction (0.08–0.12) [19,20]. UHMWPE-based materials are in great demand in mechanical engineering, where they are used as sliding bearings and protective coatings. They are also used in medicine and many other knowledge-intensive industries [21,22].

The performance and reliability of two-layer materials during operation depend significantly on the interphase interaction between the elastomer and the applied layer of the polymer. In most cases, polymers and rubber are incompatible and are regarded as a dispersed system where one component is distributed within another one [23]. In this regard, when creating a material based on UHMWPE and elastomer, it is necessary to take into account their physico-chemical properties, which affect their ability for adhesive interaction. Elastomer is a multi-component mixture of rubber-based ingredients and various fillers. The UHMWPE polymer has low surface energy and is nonpolar and chemically inert. Therefore, the formation of a strong adhesive interaction between it and other polymers is a rare occurrence. Therefore, UHMWPE and elastomers can be considered incompatible materials so that when they interact, the thermodynamic system does not reach equilibrium with minimal energy. There are well-known studies on the combination of rubber and UHMWPE. In their research paper, Kondo et al. [17] examined the influence of modified UHMWPE fibers on the properties of butadiene-styrene rubber. In order to increase the adhesion of the UHMWPE fibers to the rubber, the polymer was modified by electron-beam irradiation with subsequent polymerization of the compound with the polar group. In another research [18], a modification was carried out by preliminary ozone treatment followed by ultraviolet grafting of glycidyl methacrylate onto UHMWPE fibers. Thus, stronger adhesion can be achieved by mechanical bond, physico-chemical effects, and their combinations.

The examples given, however, refer to the volumetric modification of elastomer, and the influence of UHMWPE on surface properties is negligible. One approach to enhancing adhesion between these materials involves addition of reactive compounds that influence their chemical bond and thus their inter-molecular interaction. We have previously shown the influence of vulcanization accelerators on the compatibility of UHMWPE and elastomer [24]. It has been determined that the introduction of up to 0.3 phr diphenylguanidine into an isoprene-rubber mixture leads to an increase in adhesion strength in a two-layer material. The results show that introduction of suitable vulcanization accelerators is a reasonable and effective approach to enhancing the interfacial interaction between UHMWPE and rubber.

This paper analyzes combining an elastomer and a modified UHMWPE to obtain two-layer materials. The purpose of this research is to study the effect of 2-mercaptobenzothiazole, diphenylguanidine, and tetramethylthiuram disulfide on the properties of UHMWPE, as well as their effect on the interfacial interaction of the latter with an elastomer.

## 2. Materials and Methods

### 2.1. Loading and Testing System

Tensile strength and elongation at break were tested with the universal testing machine Autograph ASG-J (Shimadzu, Tokyo, Japan) at room temperature in accordance with the ISO 37-2020. Mechanical properties of UHMWPE-based composites were measured according to ASTM D3039/D3039M-14. Adhesion strength between the UHMWPE and the elastomer was measured with the testing machine at room temperature and the speed of grippers’ movement of 50 mm/min according to ISO 36-2021.

The linear thermal expansion coefficient of the samples was measured on the TMA-60 thermomechanical analyzer (Shimadzu, Kyoto, Japan) according to ISO 11359-2:2021. The analysis was carried out during punch penetration into a sample of 10 × 10 × 2 mm at a temperature rise of 10 °C/min in a helium medium, using liquid nitrogen as a refrigerant. The diameter of the punch was 2.5 mm, the weight on the punch was 0.50 N, the tests were conducted within a temperature range of −80 to +100 °C.

The study of the microstructure of the low-temperature chip samples was carried out on the JSM-7800F raster electron microscope (JEOL, Akishima, Japan) in the mode of secondary electrons.

The tribological properties of composites based on UHMWPE were analyzed with the help of tribometer UMT-3 (CETR, Mountain View, CA, USA) on the friction scheme «finger-disk». The mass of the samples was measured on the analytical weights Discovery DV215CD (OHAUS, Greifensee, Switzerland) with an accuracy of 0.00001 g.

The infrared absorption spectra were obtained on an IR infrared spectrometer with the Fourier transformation Varian 7000 FT-IR (Varian 7000, Palo Alto, CA, USA) in the range 500–3000 cm^−1^. Thin films of composites and two-layer materials were used in the research. The tests were carried out at a resolution of 2 cm^−1^, and the number of scans per spectrum—16 scans.

The scheme and structure of the study is shown in Figure 1.

### 2.2. Experimental Materials and Parameters

SKI-3 isoprene rubber (SIBUR, SNHZ, Nizhnekamskneftekhim, Nizhnekamsk, Russia; specification: TU 2294-037-48158319-201). SKI-3 (IR) is a synthetic isoprene rubber, which is produced by the solution polymerization of isoprene and contains at least 96% cis-1.4-links and is filled with a darkening antioxidant. The ingredients used were common for rubber mix and polymer, such as stearic acid (GOST 6484-96), 2-mercaptobenzimidazole (MBT) (GOST 739-74), zinc oxide (GOST 10262-73), sulfur (GOST 127.4-93), diphenylguanidine (DPG) (GOST 40-80), carbon black trademark K-354 (GOST 7885-86), and tetramethylthiuram disulfide (TMTD) (GOST 25127-82). UHMWPE GUR 4022 (Celanese, Nanjing, China) with a molecular mass of 5 million g/mol, with an average particle size of 145 µm and a melting point of ~130–135 °C was used.

#### Technology of Combining Two Materials

Rubber and other ingredients (stearic acid, sulfur, ZnO, DPG, MBT, and carbon black), according to the formulation given in Table 1, were mixed in a PL-2200 plasticorder (Brabender GmbH&Co., KG, Duisburg, Germany) for 20 min at initial mixing temperature of 40 °C and 30 rpm. The resulting composition is a standard rubber mixture based on SKI-3 isoprene rubber. The distinctive feature of the rubber compound is an additional introduction of the secondary accelerator DPG.

The polymer-based composite mixture was produced via dry blending of UHMWPE with a filler (MBT, DPG, and TMTD) in a blade mixer at a rotor rotation rate of 1200 rpm. The polymer composite material (PCM) contents are shown in Table 2.

The production of two-layer materials, where one layer is a UHMWPE, and the other layer is an elastomer, was carried out in four stages:Molding of the UHMWPE layer was carried out for 5 min in the mold under a pressure of 10 MPa at room temperature, the thickness of the layer for examination was ~4 mm.A rubber mixture with ~6 mm thickness that was required for the study was laid over the molded layer of the UHMWPE.The sample mold was placed in the PCMV-100 hydraulic vulcanization press (Impulse, Ivanovo, Russia), heated to +155 °C, for 20 min at 10 MPa pressure.Cooling was carried out in a mold under pressure in a hydraulic press at a temperature up to +80 °C.

A special feature of the two-layer materials produced in this way is that it is possible to adjust the thickness of the UHMWPE layer depending on the mass of the powder, which makes it possible to adjust the thickness of the elastomer from the mold used. Figure 2 shows a photograph of the UHMWPE/elastomer double-layer material.

## 3. Results and Discussion

### 3.1. Characteristics of UHMWPE-Based Composites

Figure 3 shows the results of the mechanical and tribological properties of UHMWPE and composites based on it. Modified UHMWPE with vulcanization accelerators was used to make a two-layer material. Therefore, we first studied the effect of these fillers on the mechanical and tribological properties of UHMWPE at a concentration of 0.5, 1, and 2 wt.%.

As shown in Figure 3, the introduction of fillers (MBT, DPG, TMTD) into UHMWPE results in an increase in tensile strength of 28–53% and elongation at break of 7–23%, if compared to the initial polymer. It is shown that composites containing vulcanization accelerators are characterized by high values of Young’s modulus compared to the original UHMWPE. Thus, with the introduction of vulcanization accelerators, an increase in the values of Young’s modulus by an average of 46–75% was recorded. The deformation and strength characteristics between composites do not change depending on the composition and content of the filler. This change in properties is because injected fillers facilitate relaxation processes during the application of tensile forces [19]. Thus, the organic filler acts as a plasticizer for the load-bearing structure. The increase in the strength and Young’s modulus of PCM can be explained by the increase in the stiffness of the material due to the interaction of the vulcanizers with the polymer matrix within the amorphous phase, as was shown by the IR method (Figure 4).

Based on the results of the tribological properties of the PCM, it is established that the introduction of 1 wt.% of DPG, MBT, TMTD fillers in UHMWPE reduces the mass wear rate by threefold compared to the original polymer. Some wear resistance is observed in composites with DPG and TMTD at 0.5 wt.% and UHMWPE/2 wt.% TMTD. The coefficient of friction of the PCM is independent of the type of filler and its content and remains at the level of the initial UHMWPE.

IR spectroscopy analysis was carried out to establish the interaction between fillers and UHMWPE (Figure 4). In the infrared spectra of composites, peaks in the region of 2790 and 1465 cm^−1^ correspond to the valence, and deformation vibration of CH_2_ groups, as well as peak at 716 cm^−1^ correspond to the vibration of CH_2_ polyethylene (-CH_2_-CH_2_-)_n_. The absorption bands at 2340, 2020, 1895, 1368, and 1305 cm^−1^ refer to the amorphous and crystalline regions of the UHMWPE associated with vibration (-CH_2_) groups of the polymer chain [25].

The IR spectrum of thin PCM film containing MBT and DPG, revealed the absorption bands of oxygen-containing groups in the area 1500–1645 cm^−1^, related to vibrations -C=O of bonds. IR peaks in the area 1180–1400 cm^−1^ associated with vibrations -C-O-C- bond. Absorption bands of oxygen-containing groups are known [26] to occur as a result of oxidation processes of the filler itself and the polymer matrix, i.e., when the UHMWPE is obtained by hot pressing, injectable fillers can initiate oxidative reactions in the composite mixture. IR spectra exhibit characteristic peaks related to TMTD and MBT. For example, when TMTD is introduced into the UHMWPE, it exhibits peaks corresponding to the vibration of nitrogen-containing groups (C-N binding oscillations): Primary amine at 1090–1020 cm^−1^, secondary amine at 1190–1130 cm^−1^, and tertiary amine at 1210–1150 cm^−1^. In the case of PCM filled with MBT, the IR spectrum is distinguished by the presence of peaks in the area 1340–1250 cm^−1^, corresponding to C-N coupling vibrations in aromatic compounds. There are also peaks in the 670–1225 cm^−1^ range, which correspond to the oscillations of C-H and C=C of the benzene ring bonds. At the same time, DPG-filled PCM does not have characteristic peaks of DPG functional groups, but there is an increase in the absorption bands of the carboxylic group.

Thus, the introduction of MBT, DPG, and TMTD into UHMWPE not only improves the mechanical properties but also oxidizes the polymer matrix, which can increase the interaction of the polymer with other materials.

### 3.2. Characteristics of the Elastomeric Material

The mechanical properties of the isoprene rubber elastomer are shown in Table 3. The stress-strain curve is shown in Figure 5.

As can be seen from Table 3 and Figure 5, the tensile strength of elastomeric sample 1 (sample No. 1 in Table 1) is 22 MPa, the elongation at break is 879%, the tensile stress at 100% elongation (modulus) is 1.8 MPa. With the additional introduction of TMTD into the elastomeric material (sample No. 2 in the Table 1), there is a decrease in relative elongation by 38% and tensile strength by 23%. The high modulus properties of elastomers increased by 61% with the addition of TMTD. The simultaneous use of several vulcanization accelerators (DPG, MBT, TMTD) in the rubber compound leads to an increase in the number of sulfur bonds between rubber macromolecules [27]. As a result, there is a decrease in the mobility of the polymer chain and the elasticity of the elastomeric matrix.

### 3.3. Characteristics and Structure of the Two-Layer Materials

Analysis of the adhesive strength between the layers of a two-layer material showed that the destruction of the original UHMWPE with rubber occurs according to the cohesive mechanism of delamination between the materials. In the case of two-layer materials with modified UHMWPE, cohesive breakdown occurs during delamination. Such material breakdown of elastomer/UHMWPE-DPG, elastomer/UHMWPE-MBT, and elastomer/UHMWPE-TMTD indicates cohesive destruction, while adhesion exceeds the cohesive strength of the rubber (Figure 6).

Figure 7 presents the results of analyzing the strength of the bond between elastomer and UHMWPE filled with MBT, DPG, and TMTD during delamination.

As Figure 7 shows, the original two-layer material has an elastomer/UHMWPE bond strength of 9.6 N/mm in the case of polymer delamination. With the introduction of 0.5 wt.% DPG, MBT, and TMTD into UHMWPE, an increase in adhesive strength is observed when tested for delamination relative to the original material. The introduction of MBT into UHMWPE leads to an increase in the adhesive strength between the elastomer and polymer layers by 35% relative to the original two-layer material. In the case of the elastomer/UHMWPE-DPG sample, there is a slight increase in the bond strength upon delamination, which is 11.64 N/mm. The maximum bond strength between the layers is found in the two-layer elastomer/UHMWPE-TMTD material, which is two times higher than elastomer/UHMWPE-DPG and elastomer/UHMWPE-MBT.

As noted above, when two-layer materials are laminated, the fracture occurs along the rubber, but the strength of the bond between the layers in case of delamination varies considerably. The different bond strength of cohesive breaking patterns of two-layer materials can be explained by the fact that the modifiers used in UHMWPE are accelerators of rubber vulcanization. Hence, vulcanization accelerators injected into UHMWPE during sintering interact not only with the polymer but also with rubber along the partition and contribute to increased rubber stitching density. This results in an increase in the rubber tensile module, as shown in Table 3. Therefore, the strength of the rubber and, correspondingly, the bond strength between the elastomer and the UHMWPE, filled with MBT, DPG, and TMTD during delamination, are increased.

The modification of UHMWPE through the introduction of MBT, DPG, and TMTD is the key to enhancing the adhesive interaction between rubber and UHMWPE. Figure 8 presents a schematic representation of the possible process of interaction between the rubber and UHMWPE, filled with vulcanization accelerators. The presence of vulcanization accelerators in the rubber mixture and in the UHMWPE stimulates a more active formation of sulfide bonds along the border between the UHMWPE and rubber, thus increasing the strength of the compound.

The structure of the phase boundary of the two-layer materials based on IR and UHMWPE is shown in Figure 9 (images obtained by scanning electron microscopy).

Figure 9 shows the IR rubber phase boundary with the initial UHMWPE, where the difference between polymer and rubber is clearly observed. With a large increase in the above-molecular structure of the two-layer material, penetration of individual fibrillary macromolecules into the rubber is observed, which may indicate a strong interphase interaction, resulting in increased adhesion between the materials. At the phase boundary, the UHMWPE has a fibrillary structure, which may also indicate chemical interaction and the formation of a strong adhesive compound. There is documented evidence [24] of the effect that DPG, when present in a rubber mixture, produces on the supramolecular structure of UHMWPE in the area of the interfacial boundary, due to which the adhesion between materials increases.

The introduction of vulcanization accelerators (containing 0.5 wt.% DPG, MBT, and TMTD) into the UHMWPE layer changes the structure of the UHMWPE at the interphase. It is clear that the structural formations of the fibril are visually reduced and look denser. Higher magnification shows that UHMWPE fibrillary macromolecules also penetrate the elastomer at the interfacial boundary of the two-layer materials. It is possible that the refinement of fibrillary structures at the interphase boundary results in an increased number of macromolecule-rubber bonds, indicating the formation of a strong adhesive compound and the toughening of the rubber itself due to integration of filler particles.

To assess the formation of new bonds and influence of injected fillers, IR spectra were obtained, which are shown in Figure 10.

Similarly to the IR spectra of UHMWPE-based composites, absorption bands corresponding to the deformation and valence oscillations of CH_2_ bonds, amorphous and crystalline areas of polyethylene, pendulum oscillations of the polymer chain: 720, 1368, 1465 and 1310 cm^−1^ were found. The IR sample exhibited bands of sulfide-containing compounds: sulfide bonds (S-S) at 570 cm^−1^, valence oscillations of R-SO-OR and RO-SO-OR groups at 1129 and 1240 cm^−1^. The broadening of these peaks may indicate the active interaction of sulfur with the macromolecules of rubber and UHMWPE through the sulfide groups, which influence the strength of the adhesive interaction between elastomer and polymer. The peak at 1240 cm^−1^ may also belong to the C=S and C-S coupling oscillations, which are related to TMTD. Another characteristic of IR is the vibration of nitrogen-containing groups, such as peak at 1140 cm^−1^, caused by C-N bond vibration, absorption area at 1650–1540 cm^−1^, corresponding to amine group (NH linkage fluctuations). The peak at 1661 cm^−1^ corresponds to the stretching vibrations of C=N-O groups (oximes). In addition, the specimens are characterized by vibrations of oxygen-containing groups, so the peak at 1498 cm^−1^ refers to the fluctuations of the carboxylic groups (C=O), and the absorption bands are about 1250–1100 cm^−1^ and 980–870 cm^−1^, caused by fluctuations in the C-O and O-O groups. Moreover, the absorption band in the region of 800–1000 cm^−1^ can correspond to vibrations of unsaturated C=C bonds, the so-called trans-vinylene groups, formed due to cross-linking of carbon bonds. In addition, the IR spectra revealed the presence of ether and oxo compounds, which are represented by absorption bands in the region of 890–820 cm^−1^ of the peroxide group and vibrations of the C-O-O bond. The peak at 1088 cm^−1^ is caused by stretching vibrations of the aliphatic ether-oxygen bond C-O-C [28,29,30].

The analysis of IR spectra (Figure 10) revealed that in a two-layer material containing TMTD, there is an increase in the intensity of peaks related to IR and UHMWPE. At the same time, there is broadening of the absorption bands of sulfide and ether bridge bonds, which implies the formation of cross-linked structures between the two materials.

The use of two-layer elastomer and UHMWPE-based material can cause large temperature fluctuations. Therefore, a mismatch of linear extensions may lead to tension in the interfacial region. Consequently, a possibly significant change in the initial linear dimensions of the product may cause the two-layer material to break down. Thus, a clear understanding of the changes in linear dimensions at different temperatures is key to the study of the thermal stability of a two-layer material. Figure 11 shows the results of analyzing the thermomechanical curves of the original UHMWPE, reinforced UHMWPE, and elastomer.

Figure 11 shows that during thermal expansion in the temperature range from −80 °C to +100 °C, the change in linear dimensions for the original UHMWPE is 5%, UHMWPE + 0.5 wt.% MBT is 3.3%, UHMWPE + 0.5 wt.% DPG is 3.2%, UHMWPE + 0.5 wt.% TMTD is 2.9%, and for rubber, based on IR, the change in linear dimensions is 0.7%. The difference in the temperature dependence on deformation of the rubber IR between the original UHMWPE and the enhanced one is from 2.2% to 4.3%, which with frequent and strong ambient temperature variations, can lead to the destruction of two-layer products due to the irregular pattern of change in linear dimensions. The smallest difference of 2.2% in terms of linear dimensions’ change was observed between the TMTD-filled UHMWPE and elastomer.

## 4. Conclusions

Based on the conducted research, the following conclusions can be drawn:The introduction of TMTD, DPG, and MBT into the UHMWPE layer leads to an increase in adhesion between UHMWPE and rubber. The greatest increase in adhesion between rubber and UHMWPE occurs with the introduction of TMTD, which is up to two times higher compared to the materials of the compositions elastomer/UHMWPE-DPG and elastomer/UHMWPE-MBT.It has been established that the increase in adhesion between UHMWPE and elastomer is due to the chemical nature of functional additives containing reaction centers that provide chemical cross-linking between the components of the two-layer material. In addition, the oxidative processes that occur during the processing of polymers also contribute to the appearance of oxygen-containing groups involved in intermolecular interaction.The SEM method shows the formation of a dense connection between polymers at the interface of a two-layer material and a change in the supramolecular structure of UHMWPE during the introduction of DPG, MBT, TMTD to a denser fibrillar structure.The study of linear thermal expansion showed that the introduction of MBT, DPG, TMTD into UHMWPE reduces the linear expansion in the temperature range from minus 80 °C to plus 100 °C. Temperature changes in the linear dimensions of UHMWPE and composites with 0.5 wt.% MBT, DPG, TMTD from 2.9% to 5%, the change in rubber based on the IR is 0.7%.The developed materials are designed to produce products that have, on the one hand, high strength and antifriction properties (UHMWPE) and on the other hand, damping properties, elasticity, and resistance to fatigue (rubber).

In further research, the authors plan to supplement the studies on the determination of the cross-link density between UHMWPE and IR, using data from measurements of rheological properties. We also plan to evaluate the degree of cross-linking of a two-layer material using nuclear magnetic resonance spectroscopy on the 13C nucleus, in which the mechanism for the formation of branched macromolecules with different architectures is established, and the degree of branching is determined.

## Figures and Tables

**Figure 1 materials-15-04678-f001:**
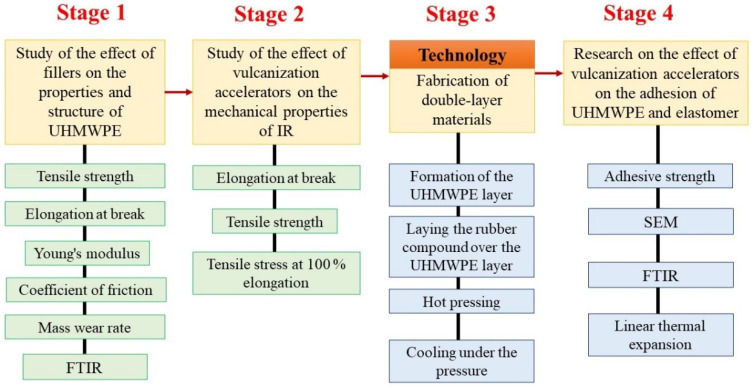
Scheme of the experiment.

**Figure 2 materials-15-04678-f002:**
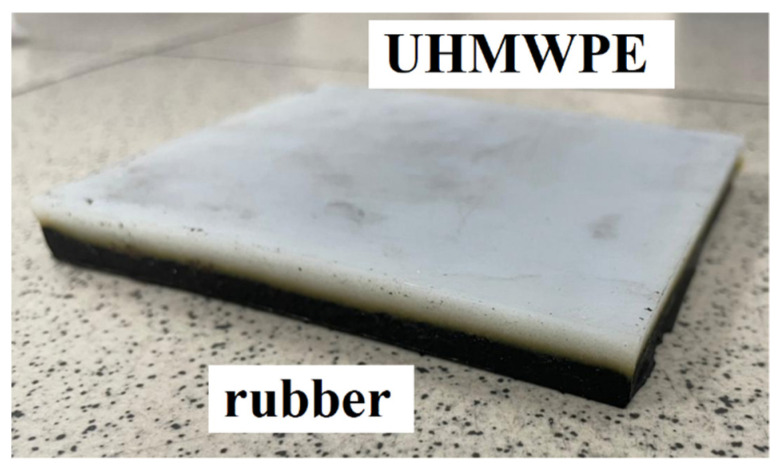
Double layer UHMWPE/elastomer material.

**Figure 3 materials-15-04678-f003:**
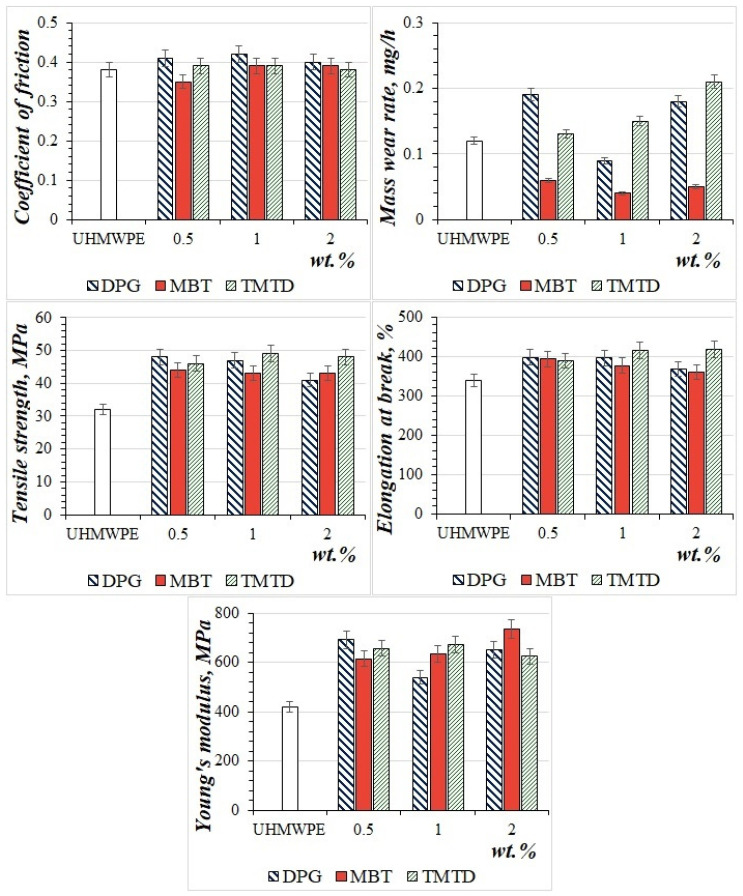
Deformation-strength and tribological properties of UHMWPE and PCM.

**Figure 4 materials-15-04678-f004:**
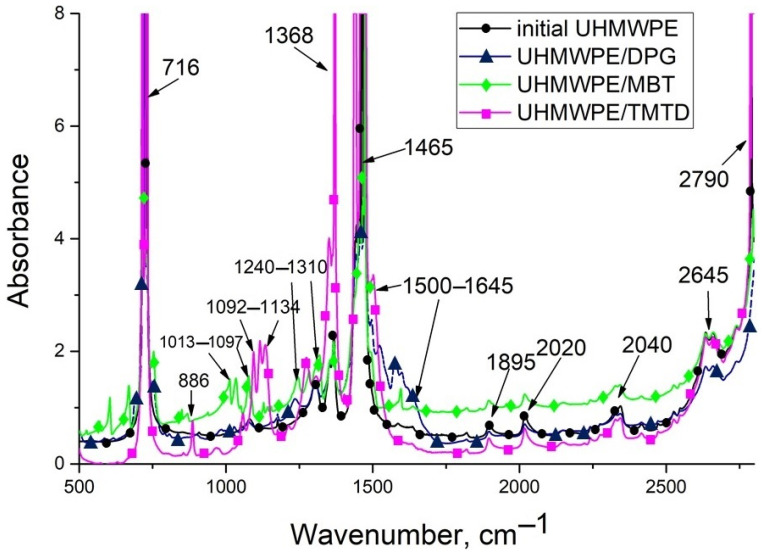
FTIR spectra of PCM based on UHMWPE.

**Figure 5 materials-15-04678-f005:**
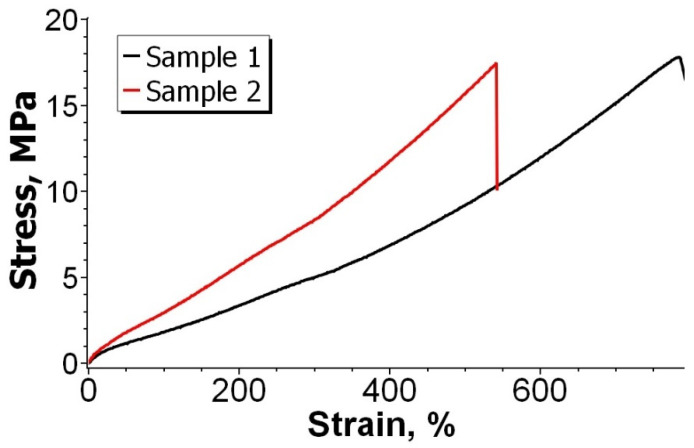
Stress-strain curve of the tensile tests.

**Figure 6 materials-15-04678-f006:**
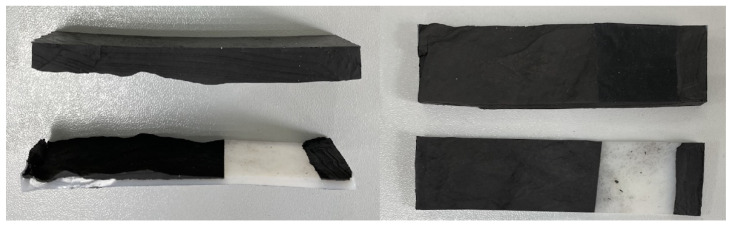
Cohesive nature of delamination of two-layer materials (e.g.,: elastomer/UHMWPE-TMTD).

**Figure 7 materials-15-04678-f007:**
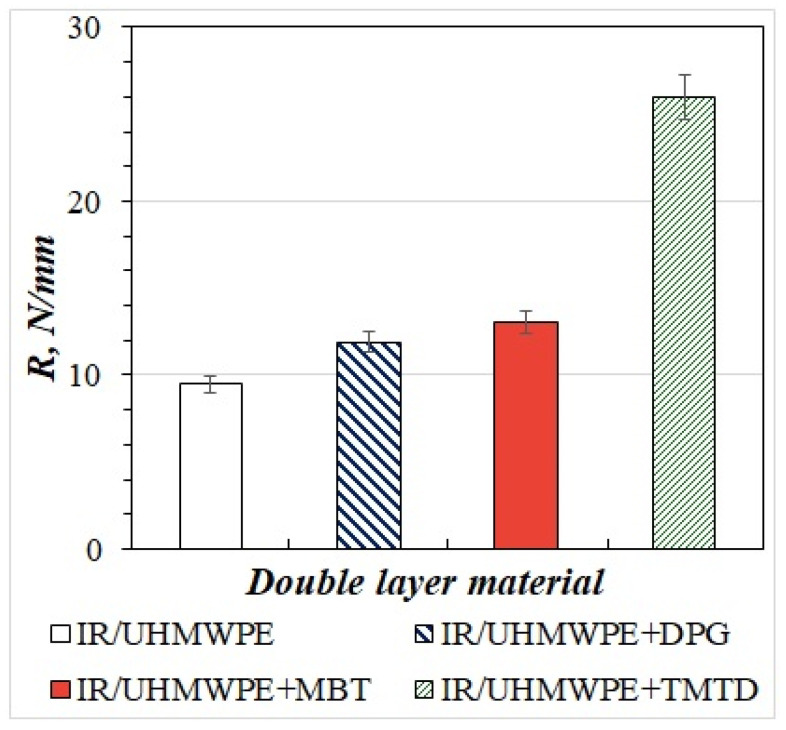
Dependence of bond strength between elastomer and UHMWPE on the content of DPG, MBT and TMTD.

**Figure 8 materials-15-04678-f008:**
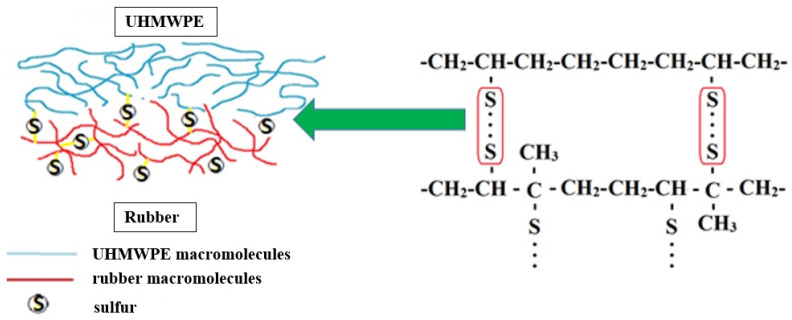
Dependence of bond strength between elastomer and UHMWPE on the content of DPG, MBT, and TMTD.

**Figure 9 materials-15-04678-f009:**
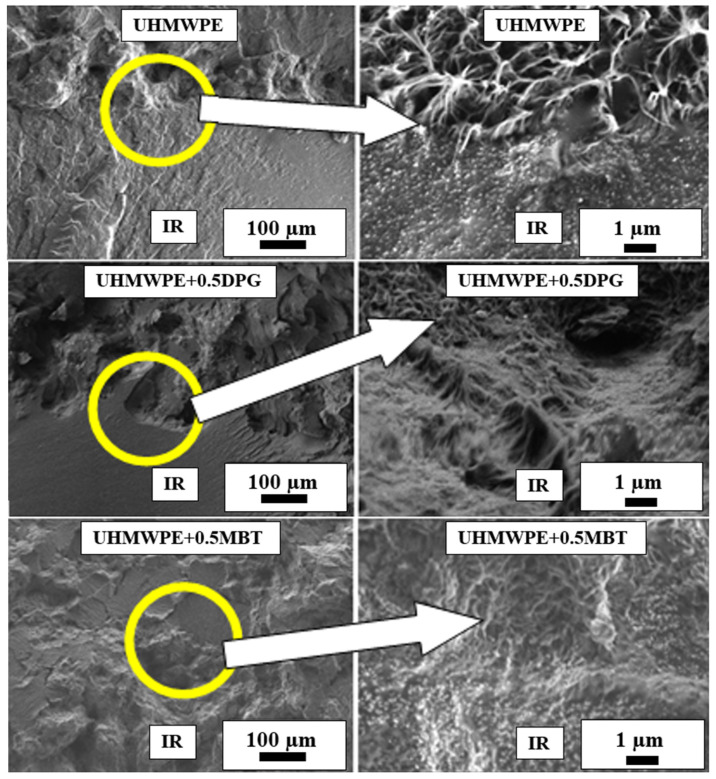
Phase boundary of interaction between IR rubber and UHMWPE.

**Figure 10 materials-15-04678-f010:**
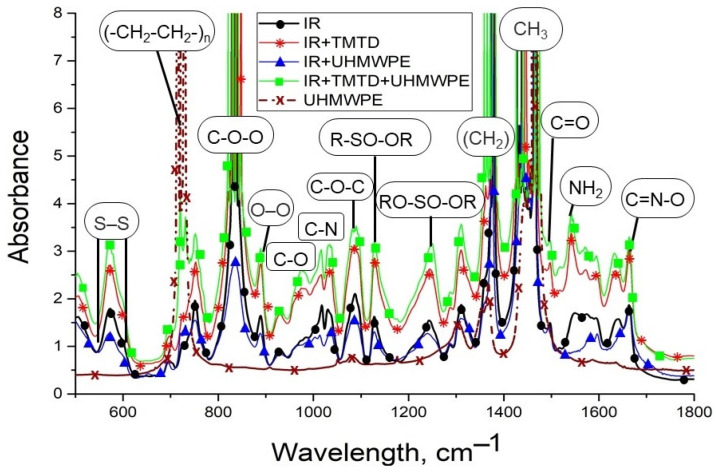
Comparison of IR spectra of initial polymer materials and UHMWPE mixture with elastomer.

**Figure 11 materials-15-04678-f011:**
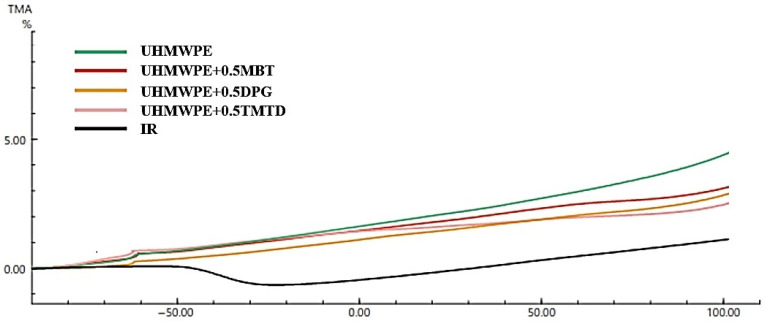
Linear thermal expansion of UHMWPE, UHMWPE + 0.5 wt.% DPG, UHMWPE + 0.5 wt.% MBT, UHMWPE + 0.5 wt.% TMTD and rubber based on IR rubber in the temperature range from −80 °C up to +100 °C.

**Table 1 materials-15-04678-t001:** Isoprene Rubber Standard Blend Formulation (Formulation: Bulk Parts per 100 Rubber Parts).

No.	Compounds	phr		Time of Introduction, min
		1	2	
1	IR	100.0	100.0	0
2	Stearic acid	2.0	2.0	0
3	2-mercaptobenzotiazole	1.5	1.5	10
4	Zinc oxide	5.0	5.0	5
5	Sulfur	2.0	2.0	15
6	Diphenylguanidine	0.3	0.3	10
7	TMTD	-	0.5	10
8	Carbon K-354	50.0	50.0	2

**Table 2 materials-15-04678-t002:** The content of fillers in polymer composite materials based on UHMWPE.

Composite Number	Composition, wt.%			
	UHMWPE	DPG	MBT	TMTD
1	99.5	0.5	-	-
2	99.5	-	0.5	-
3	99.5	-	-	0.5

**Table 3 materials-15-04678-t003:** Mechanical properties of the elastomeric materials.

Samples	Elongation at Break ε_p_, %	Tensile Strength f_p_, MPa	Tensile Stress at 100% Elongation f_100_, MPa
1	879 ± 50	22 ± 1	1.8
2	539 ± 30	17 ± 1	2.9

## Data Availability

The data used to support the findings of this study are available from the corresponding author upon request.

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
