# Peer review of "Two-Layer Rubber-Based Composite Material and UHMWPE with High Wear Resistance"

_materials, 2022, doi:10.3390/ma15134678_

Round 1

Reviewer 1 Report

1.     The authors state that UHMWPE stands for inconsistently, such as ultra-high-molecular-weight polyethylene in line 15-16 and ultra-high molecular weight polyethylene in line 57-58. Please revise it.

2.     Alphabetic order in the abstract is needed, please change it.

3.     In line 30-31, elastomeric materials are also adopted in medical implants, especially bearing components on the total hip prosthesis. Please include this important point in this sentence. Also, the authors need to adopt suggested references published by MDPI as follows: Computational Contact Pressure Prediction of CoCrMo, SS 316L and Ti6Al4V Femoral Head against UHMWPE Acetabular Cup under Gait Cycle. J. Funct. Biomater. 2022, 13, 64. https://doi.org/10.3390/jfb13020064

4.     The present study is lack novelty. After the reviewer evaluation, nothing something really new was found. Can the authors explain it? Highlight the study’s novelty in advance if it has.

5.     There are several elastomeric materials other than UHMWPE. But why UHMWPE is selected rather than other elastomeric materials. It is needed to explain in the introduction section.

6.     The research flow needs to be explained in the materials and methods section with an additional illustrative figure to make the reader more interested and easier to understand rather than specific figures and dominant words.

7.     Uppercase and lowercase of the subsection should be revised based on the MDPI format.

8.     Comparison with the previous study needs to be stated in the results section. The present study discusses UHMWPE with high wear resistance. Wear is related to the mechanical properties of materials. The authors need to explain it in the introduction and/or discussion section. Please include it. The suggested reference before would be adopted.

9.     Related to the mechanical properties of materials, the authors have been evaluating UHMWPE, DPG, MBT, and MTMD. But the authors do not seek the difference of young modulus and poison ratio as mechanical properties of the materials tested. It is needed to include.

10.  Research limitation needs to be explained before the conclusion section.

11.  The present conclusion is not concise, please revise it.

12.  Further research needs to explain in the conclusion section.

13.  Non-published materials are not written in English, please change the language.

14.  Please revised the error used in grammar, punctuation, and style of the English used in the whole manuscript.

Reviewer 2 Report

The manuscript entitled „Two-layer Rubber-based Composite Material and UHMWPE with High Wear Resistance” describes the development of two-layer composite materials produced from ultra-high-molecular-weight polyethylene (UHMWPE) and isoprene rubber. In order to obtain improved adhesion and performance of the two-layer materials, a series of accelerators, such as 2-mercaptobenzothiazole, diphenylguanidine and tetramethylthiuram disulfide have been applied. The manuscript has numerus flaws related to the terminology used and information’s presented. There are missing information related to methodology and information for the results. However, the work is interesting and might be useful to the readers of materials journal, but there are some points need to be corrected in the manuscript:

1. lines 35-40 and 50-56 (introduction) should be referenced to suitable literature.

2. lines 101-106 (tensile and adhesion tests methodology) - how many samples were used for measurement? Has a statistical analysis of the results been carried out? What was the thickness of the samples?

3. lines 120-123 (FTIR methodology) - how many scans? What resolution? Was baseline correction performed?

4. Why the authors selected 50 phr of carbon black concentration in the study? Please explain it.

5. lines 169-170 – the authors mentioned in the text : “To assess the effect of modifiers on the properties of the UHMWPE polymer matrix, fillers were introduced at concentrations of 0.5, 1, and 2 wt.%....”. In my opinion this sentence is inconsistent with the nomenclature used in elastomer technology, because the compounds studied in the work belong to the group of (cross-linking) accelerators. Please change the misleading words or sentences throughout the manuscript.

6. Figure 2a - It can be seen that a very low concentration of DPG or MBT (0.5 phr) incorporated into UHMPWE causes a very significant increase in tensile strength (even over 50%, the authors claim). Have the authors encountered such an improvement in TS after introducing such additives to UHMPWE in the literature reports? Please explain it and compare with literature.

7. lines 214-224 – please improve the discussion on the mechanical performance of elastomers and add stress-strain curves for better understanding their tensile properties.

8. lines 250-267 - If the cross-linking accelerators included in UHMWPE can affect the mechanical properties of the rubber after preparation of two-layered material, it would also be worth testing whether they have impact on the crosslink density and curing process. Therefore, I suggest performing additional rheometric measurements and crosslink density analysis for the rubber fragments in contact with UHMWPE, and then compare the effect of accelerators.

9. Please correct the Figures style according to the Materials journal guidance.

10. lines 339-348- why the smallest difference in terms of linear dimensions change was observed for composite containing TMTD? Please explain it.

11. Conclusion – the authors mentioned: “The developed materials are intended for manufacturing of products that, on the one hand, have high strength and antifriction properties  (UHMWPE), and, on the other hand, have damping properties, elasticity and resistance to fatigue (rubber).” It is a very general statement. Please provide a particular example of materials where the studied two-layer composites could find application.

Round 2

Reviewer 1 Report

This manuscript is recommended for publication.

Reviewer 2 Report

The authors have improved their manuscript following the Reviewer's guidelines.